# Investigating Additive Manufacturing Possibilities for an Unmanned Aerial Vehicle with Polymeric Materials

**DOI:** 10.3390/polym16182600

**Published:** 2024-09-14

**Authors:** Laura Šostakaitė, Edvardas Šapranauskas, Darius Rudinskas, Arvydas Rimkus, Viktor Gribniak

**Affiliations:** 1Department of Aeronautical Engineering, Vilnius Gediminas Technical University (VILNIUS TECH), Linkmenų Str. 28-4, 08217 Vilnius, Lithuania; laura.sostakaite@stud.vilniustech.lt (L.Š.); edvardas.sapranauskas@stud.vilniustech.lt (E.Š.); darius.rudinskas@vilniustech.lt (D.R.); 2Laboratory of Innovative Building Structures, Vilnius Gediminas Technical University (VILNIUS TECH), Sauletekio Av. 11, 10223 Vilnius, Lithuania; arvydas.rimkus@vilniustech.lt

**Keywords:** 3D printing, polylactic acid (PLA), prototyping, mechanical performance, bending test

## Abstract

Fused filament fabrication, also known as fused deposition modeling and 3D printing, is the most common additive manufacturing technology due to its cost-effectiveness and customization flexibility compared to existing alternatives. It may revolutionize unmanned aerial vehicle (UAV) design and fabrication. Therefore, this study hypothesizes the 3D printing possibility of UAV using a simple desktop printer and polymeric material. The extensive literature analysis identified the acceptable prototyping object and polymeric material. Thus, the research focuses on applying polylactic acid (PLA) in manufacturing the flying wing-type UAV and develops a fabrication concept to replicate arial vehicles initially produced from a mixture of expanded polystyrene and polyethylene. The material choice stems from PLA’s non-toxicity, ease of fabrication, and cost-effectiveness. Alongside ordinary PLA, this study includes lightweight PLA to investigate the mechanical performance of this advanced material, which changes its density depending on the printing temperature. This proof-of-concept study explores the mechanical properties of printed parts of the wing prototype. It also considers the possibility of fragmentation in fabricated objects because of the limitations of printing space. The simplified bending tests identified significant reserves in the mechanical performance regarding the theoretical resistance of the material in the wing prototype, which proves the raised hypothesis and delivers the object for further optimization. Focusing on the mechanical resistance, this study ignored rheology and durability issues, which require additional investigations. Fabricating the wing of the exact geometry reveals acceptable precision of the 3D printing processes but highlights the problematic technology issues requiring further resolution.

## 1. Introduction

Following the American Society for Testing and Materials (ASTM) description [1], additive manufacturing (AM) involves depositing material layer by layer based on digital 3D design data to create physical objects. The well-known terminology “3D printing”, interchangeably with AM, refers to a different process; the terms rapid printing or prototyping do not also accurately describe AM technologies. A more precise AM description is a continuous material addition process, which sets it apart from conventional manufacturing methods that rely on material removal [2]. Different AM technologies exist [3], e.g., fused filament fabrication (FFF), stereolithography (SLA), digital light processing (DLP), selective laser sintering (SLS), multi-jet modeling (MJM), selective laser melting (SLM), laminated object manufacturing (LOM), and electronic beam melting (EBM). Of these techniques, FFF (also known as fused deposition modeling, or FDM) is the most common due to its cost-effectiveness and acceptable surface finish [4]. It describes the analysis object of this study. However, this manuscript ignores the above terminology issue and uses the 3D printing term to describe the FFF process. In recent years, 3D printing has emerged as the predominant method for polymer-based component manufacturing in hobby or engineering applications. This transformative technology empowers engineers to swiftly prototype and fabricate intricate geometries that were once deemed unattainable, costly, or even impossible using traditional methods. It excels in creating parts with complex internal structures or geometries. This attainment is challenging or nearly impossible using injection molding or CNC machining [5].

Modern slicing software, designed to prepare numerical models for 3D printing, features user-friendly interfaces and intuitive designs that typically do not require any prior preparation or deep technical knowledge. However, 3D printing technology has drawbacks; geometry and material characteristics can vary significantly from part to part, especially when manufacturing large-size components [6]. Dimitrov et al. [7] reported that the printing tolerance differs in different building directions and may exceed 1.5 mm in the normal direction to the printing base; the circular opening diameter error may reach 1.5%. In addition, shrinkage issues substantially compromise dimensional accuracy in printed polymeric parts. Specifically, due to thermal contraction and cooling regime variations, the thermoplastic polymer’s shrinkage rate may exceed 2% [8]. The composition of the raw material (filament or pellets) and the printing process parameters also affect the quality of the results. The primary matrix material, additives, and raw material fabrication technology can alter the 3D printing quality and mechanical properties [9,10,11,12,13,14]. Even for the raw materials from the same factory, the materials characterization process affects the print quality assessment results [15,16,17,18]. Moreover, incomplete trials using only a few parameters can produce inaccurate data about the production process, as each parameter plays a crucial role in determining the nature and quality of the final product [19,20]. When selecting safety margins, these variations between individual components can significantly affect the final result [21,22]. These aspects require consideration when the final parts are intended to be functional.

Even with the above limitations, an extensive array of materials, new and emerging technologies, or simply the slow and steady improvement of manufacturing equipment has allowed various industries to adopt additive manufacturing as one of the available technologies in their respective fields. Aviation, a conservative field due to extensive safety regulations governing the industry, has also started to adopt AM as one of the available technologies [23,24,25,26]. The main focus of aerospace components is meeting comprehensive requirements, emphasizing lightweight construction and superior mechanical performance. Still, the build and print orientations significantly affect the manufacturing quality of unmanned aerial vehicle (UAV) components [27]. Ravindrababu et al. [28] evaluated the effects of build and print orientations on the 3D printed UAV components. The mechanical properties of the printed parts were assessed by comparing three build setups (edge-up, face-up, and straight-up) and print orientations (0, 45°, and 90°). The results showed the highest stiffness and tensile strength of the edge-up samples. In addition, the build orientation possessed a more significant influence on the elastic deformation of the printed components than the pathway orientation.

Based on the small UAV framework, Azarov et al. [29] provided a 3D printing method of continuous fiber-reinforced composite (CFRC) when the frame comprises continuous carbon fibers and thermoset and thermoplastic matrix materials. The proposed technology demonstrated excellent merits in rapidly prototyping, reducing the material density, and enhancing the mechanical performance. However, continuous reinforcement typically requires a non-interruptive pathway, and the fibers’ misalignment may reduce the theoretical stiffness and ultimate resistance [2]. Karkun and Dharmalingam [30] determined that PLA (polylactic acid) and ABS (acrylonitrile butadiene styrene) filaments are the most commonly used in aircraft product applications.

As 3D printing technologies mature and become more widely adopted, they are increasingly used in small-scale applications such as the production of UAVs. The literature analysis has identified several positive and negative aspects of 3D printing in the aircraft industry compared to existing manufacturing technologies. Figure 1 schematically depicts these features. Following the references [30,31,32,33,34], 3D printing technologies substantially improve the customized fabrication process, reducing manufacturing/machining time and staffing demands. These technologies ensure flexible structure optimization, significantly reducing printed structures’ overall weight and preserving their mechanical performance. They also enable the formation of channels, microfluidics, batteries, scaffold supports, and embedded electronics. Assembly fragmentation into several components or parts may simplify the fabrication process. Alternatively, several parts can be consolidated into one object to reduce assembly steps and manufacturing costs. Using different colors of raw materials for the model parts can potentially eliminate the need to coat or paint the printed objects. The efficient use of raw materials, the inseparable feature of AM processes [2], and simplifying the processing of surplus materials for reuse and utilization ensure fabrication sustainability [35].

In addition to the advantages of the technology mentioned above, 3D printing makes the fabrication of metamaterials with conductive properties possible using embedded carbon nanotubes. It also decreases the need for wiring on the aircraft and enables the integration of electronic components such as gyroscopes, accelerometers, barometers, and GPS devices within the frame; auxetic structures, representing metamaterials with a negative Poisson ratio, exhibit controlled energy absorption and insulation properties [34].

However, as Figure 1 shows, aircraft engineering faces 3D printing drawbacks [27,30,34,35,36,37]. In particular, this fabrication process is relatively slow. The near-net shape and rough surface finishing may partly compensate for this limitation. Still, this solution is partial since finishing is required due to fatigue problems or appearance requirements. Furthermore, the current scientific level of knowledge is relatively low—most research is limited to proof-of-concept work. The 3D printed objects are also prone to material defects and geometrical inaccuracy, which require additional verifications; inspecting complex cavities, small holes, and channels is complicated and typically involves scanning and non-destructive testing. Alongside the technological problems, a lack of a reliable database of materials’ characteristics and fabrication parameters complicates the optimization of load-resistant components, especially for specific demands (e.g., fatigue, aging, creep) and reliability issues.

However, the product’s size limitation (the typical printer has a size limitation of 350 mm in diameter and 380 mm in height) determines the essential restriction of 3D printing technologies in aircraft engineering [30,34,37]. Printing dimensions are slowly increasing because of the drastic increase in equipment costs. Advanced manufacturing technologies (e.g., continuous reinforcement, multi-materials, and high-temperature resistant materials) face limitations for the same reason.

A fragmentation technique can overcome the printing space limitation [38]. Although this technology involves the printed parts’ additional (connecting) process, it efficiently extends the dimension limits of the 3D printed structures. Tiwary et al. [38] identified the following primary connection techniques: mechanical interlocking, fastening, welding (fusing), and adhesive joining. The latter method was classified as the most common connection method for 3D printed parts. Therefore, only this technique defines the discussion object in this review.

The adhesive connection process offers numerous benefits, including superior thermal and electrical insulation at the adhesive layer, good damping properties, lightweight, and tightness [39,40]. However, extensive surface preparation of the 3D printed fragments to be joined is a significant drawback. Elastic mismatch (causing stress peaks when joining dissimilar 3D printed parts), adhesives filling the holes and jamming them, and difficulty in reliable non-destructive testing (an adhesively joined part cannot be disassembled without damage) determine other adhesive joining disadvantages [41,42,43].

Despite the limitations above, Fu et al. [27] have predicted a promising future for 3D printing, a pivotal technology in the future of UAVs and other industries. This industrial transformation underscores the need for further research to explore a broader range of materials suitable for mass production. It also highlights the importance of fine-tuning machining parameters to reduce surface porosity and designing support structures to minimize a negative impact on part quality.

Most literature investigates the AM of aircraft’s mechanical parts and internal components, e.g., [23,24,29,32,33], or idealized laboratory specimens, e.g., [2,28,39,40,41]. Olasek and Wiklak [44] investigated the aerodynamic performance of the standard NACA0018 airfoil fabricated by MJM, SLS, and FFF techniques, though manufactured on a small scale (100 mm chord and 170 mm span). This study demonstrated the potential of 3D printing technology in manufacturing thin-walled and tough components for aerodynamic tests. In this context, Udroiu [45] noticed the necessity of systematic procedures for surface roughness quantification of AM materials used in aerodynamic studies, still considering small profile fragments. Szafran and Jeziorek [25] reached similar conclusions, manufacturing relatively small fuselage and wing fragments with customized layering software. Tiwary et al. [38] demonstrated the possibility of UAV adhesive fragmentation, focusing on the shape and geometry but, as [25], ignoring the mechanical performance of the printed structures.

Accounting for the above limitations and the literature review results, this manuscript investigates the possibility of manufacturing UAVs using a simple desktop 3D printer and polymeric materials. It explores the possible stages of computer-aided design (CAD) production of polymeric UAVs. A UAV prototype made from a blend of expanded polystyrene and polyethylene (EPO) is used to determine the aerial vehicle’s surface shape, dimensions, and weight to demonstrate the feasibility of the proposed design approach. This fabrication technology can optimize internal UAV space and structure (Figure 1) and ensure flexible customizing of the CAD model and polymeric aerial vehicle. This manufacturing may also reduce the staffing demands since one technician can operate several printers, and UAV fragmentation ensures the replacement of damaged components and makes the aerial vehicle’s maintenance efficient. This study carefully selects the printing material and investigates the mechanical resistance of the 3D printed parts, including those fabricated via the fragmentation technique. The focus is on the printing parameters and performance, such as fabrication speed and product tolerances. The mechanical (three-point bending) tests investigate the effects of the printing technology, temperature, speed, layer orientation, and thickness on the load-bearing capacity and stiffness of the printed parts and verify their ability to reach the theoretical component resistance.

## 2. Materials and Research Methodology

All manufacturing and experimental work was conducted in the Laboratory of Innovative Building Structures at VILNIUS TECH. This research targets the application of a simple 3D printer and polymeric materials to the manufacturing of a UAV. So, the investigation considers two typical desktop printers available in the laboratory: a Prusa i3 MK3 printer with PrusaSlicer 2.3.3 slicing software (both created by Prusa Research, Prague, Czech Republic) and Creality CR-30 3DPrintMill with Creality Slicer 4.8.2 software (both produced by Shenzhen Creality 3D Technology Co., Shenzhen, China). These apparatuses employ polymeric filaments for AM, and this condition determines the raw material form.

### 2.1. The Raw Material Market Analysis

The literature analysis (Section 1) revealed that PLA and ABS filaments are typical for UAV fabrication [30], defining the preliminary material choice for this study. Still, extensively analyzing the market and specific literature determines the final selection. Table 1 shows the analysis results collected from manufacturers’ datasheets and technical literature [46,47,48,49,50,51,52,53,54,55,56,57,58,59,60,61,62,63,64,65,66,67,68,69,70,71,72,73,74,75,76,77,78,79,80,81,82,83,84,85,86,87,88,89,90], focusing on the characteristics essential for UAV manufacturing; the terms “temperature deformation” and “deformation instability” determine the materials’ deformation behavior during printing and hardening of the polymeric material. This table mainly provides the raw filaments’ mechanical characteristics since the properties of the printed materials depend on the printing settings. Still, some characteristic numbers are included in Table 1. Singh et al. [61] and Taib et al. [63] reported a triple reduction in the tension strength of PLA regarding the minimal values obtained for raw filaments. Kodali et al. [77] found a similar decrease in mechanical strength when printing a one-layer polycarbonate film. Shrinkage may also vary depending on the fabrication settings. So, Jipa et al. [55] reported ten times lower deformations than alternative references (e.g., [50,61,64,71]); however, even reduced, the 0.3–0.5% shrinkage deformations may substantially alter the shape of the printed parts and induce residual stresses in the material. These results proclaim the necessity of preliminary fabrication of 3D printed parts, as this study, to adequately characterize those performances since they may differ from the raw material properties.

Remarkably, this study intends to use the most primitive printing equipment possible. However, Mohammadian and Nasirzadeh [57] suggested PLA choice regarding ABS because of asthma risks and reduced carbon gas emissions. Therefore, the toxic gas emissions during 3D printing describe an essential condition of this study’s material choice to avoid additional ventilation, and the analysis focuses on non-toxic materials.

Table 1 provides prices of the raw filaments collected in July 2024 for the Vilnius (Lithuania) delivery place. Following this criterion, PEI is almost ten times more expensive than other non-toxic alternatives, which makes it unacceptable for developing relatively cheap UAVs, notwithstanding its exceptional mechanical performance.

Considering the remaining alternatives in Table 1, Kopar and Yildiz [52] recommended PLA for 3D printing after comparing its mechanical performance to PETG. Holcomb et al. [71] highlighted the easier printability of PLA compared to PETG. Therefore, this study selects PLA for UAV fabrication. In addition, for comparison purposes, this work considers LW-PLA an efficient, lightweight alternative to PLA in aircraft design [91].

### 2.2. The Fabrication Object

This study develops a fabrication concept to replicate an EPO aerial vehicle using 3D printing technology. The following conditions determine the modeling object choice [92]:The valid payload should fit the 1 kg to 2 kg range to extend its application abilities.The proposed modification must empty as much space inside the UAV prototype as possible to substantiate the EPO replacement with a polymeric shell.The aerial vehicle should have minimal moving parts to reduce the number of servo mechanisms and increase the design reliability.

Analysis of the available prototypes identified the Skywalker X8 [93] as the fabrication object. Figure 2 provides a view of the fabrication prototype. Skywalker is a flying wing type of UAV. Flying wings are specially designed to maintain stability and control without vertical stabilizers. In this model, the only control surfaces are the wing ailerons; in this way, only two servo mechanisms are required per UAV. The manufacturer specified the following technical characteristics of the UAV [93]: wingspan = 2120 mm; fuselage length = 790 mm; wing area = 8 × 10^5^ mm^2^; flying weight = 2.5–3.0 kg; valid payload = 1–2 kg; gravity center = from head to back 430–440 mm; air speed = 65–70 km/h; maximum flying time = 25 min; maximum flying level = 200 m. The Skywalker X8 is manufactured from EPO, a lightweight material; the original construction consists of a solid wing and thick-shelled fuselage to maintain adequate mechanical properties. The proposed redesign of this UAV to be manufactured from polymeric material will empty the wings, which were solid before, and add some additional space to the fuselage.

A 3D scan of the Skywalker X8 created an editable CAD model to measure unknown geometrical values and determine the wing airfoil. First, the root profile of the wing was traced to obtain the root airfoil coordinates. The rough surface of the 3D scan was virtually cut, and the identified profile was smoothed to clarify the airfoil shape. The traced coordinates were uploaded to the Internet site [94], where the airfoil was recognized as Seilig S5010. The airfoil coordinate file (with “.dat”-extension) was downloaded and imported to the Solidworks (2023 Student Edition, Dassault Systèmes, Paris, France) application to begin sketching. Geometric values from the 3D scan were measured and adapted for the Solidworks sketch, which is suitable for creating the CAD model.

### 2.3. The Analysis Concept

The re-design assumes the weight equality of the UAV EPO prototype and 3D printed model. So, accounting for the takeoff weight (3.0 kg) specified by the developer [93] and assessing the UAV components’ weight as 1.2 kg defines the target 1.8 kg net PLA mass. The previous studies [2,95,96] determined the 1240 kg/m^3^ density of the printed PLA objects for the same raw material and printing setting used in this study. Thus, assuming the 1.8 kg weight of the printed fuselage and wings and knowing the surface area of the CAD model (338.8 × 10^4^ mm^2^, Section 2.2) result in the approximate 1.3 mm thickness of the printed surface under the simplified assumption that no additional walls, supports, and stiffeners are fabricated inside the UAV. This simplification is possible since this study investigates the general possibility of EPO replacement with the 3D printed polymeric material. The UAV geometry will be further optimized (reducing the wall thickness where possible) to form the internal vehicle structure if mechanical tests in this study define sufficient resistance of the 3D printed components.

This study considers the wing and fuselage connection a representative analysis point because of the substantial bending moments generated in these joints. The theoretical analysis defines the required performance of the polymeric shell, and the simplified mechanical test determines the actual resistance of the printed parts. In addition, this study investigated the possibility of fragmentation of printed components, printing the simplified bending specimens, and the exact geometry of the wing prototype.

#### 2.3.1. The Theoretical Analysis of the Wing Bending Resistance

Certification specifications for UAVs have yet to be created. Therefore, this study employs the 1.5 safety factor provided by the Joint Authorities for Rulemaking on Unmanned Systems (JARUS) [97]. This analysis assumes normal flight conditions, i.e., the load coefficient equals unity. Comparing the theoretical and experimental results will determine the actual safety margins of the printed components.

This analysis considers a wing a cantilever system subjected to distributed gravity load and aerodynamic forces. It employs the developed Solidworks CAD model to estimate the wing’s weight and geometry properties, as Figure 3 shows, which results in the 0.764 kg mass. The interested reader finds the calculation details in the available literature (e.g., [91,98]). The iterative analysis results in the maximum stress *σ_max_* = 369 kPa acting in the polymeric material in the wing joint.

#### 2.3.2. The Simplified Mechanical Test

For two reasons, this proof-of-concept experimental study replaced the cantilever test with the simplified three-point bending test and simplified specimen geometry. First, typical wing mechanical tests, e.g., [99], employ elaborate testing setups because of the variable wing geometry and uniform load distribution. In addition, a limited restraint stiffness is characteristic of printed polymeric structures [100], which substantially complicates forming the cantilever support, representing the wing and fuselage joint (Figure 3). Second, the existing printing space limits the sample dimensions, which can be fabricated without fragmentation. Therefore, this study transforms the wing into an equivalent rectangular shape, which forms the prismatic sample for the three-point bending test. This test ensures the precise control of the support and loading conditions and captures the deformation and failure of the polymeric element. The bending specimen has the average height of the wing. The width of the prismatic specimen was set arbitrarily, accounting for the limitations of the printers. This specimen transformation results in 150 mm × 18.8 mm × 450 mm prismatic geometry. The prismatic specimen assumes the 1.3 mm wall thickness determined in this section above.

The Creality CR-30 3DPrintMill fabricated this specimen with 100% infill of the 1.3 mm shell (Figure 4). No internal infill was used in this sample, resulting in the 213 g weight. The fabrication assumes a 45° inclination of the printing pathway. This manufacturing layout has no nominal length limitations, so it was used for single-part wing prototyping. The following settings were used: extruder temperature = 210 °C; print bed temperature = 60 °C; print speed = 40 mm/s; layer height = 0.2 mm; nozzle width = 0.4 mm.

Apart from ordinary PLA, this study includes two bending samples printed from lightweight PLA (LW-PLA) to investigate the mechanical performance of this advanced material, which changes its density depending on the fabrication temperature (Section 2.1). The following settings were used for LW-PLA to ensure a 20% weight reduction, raising the nozzle and bed temperature as suggested in [101]: extruder temperature = 230 °C; print bed temperature = 70 °C; print speed = 35 mm/s; layer height = 0.24 mm; nozzle width = 0.4 mm. Figure 5 shows the fabrication process of both polymeric specimens.

This study also considers the possibility of wing fragmentation, accounting for the printing-space limitations of a typical apparatus, e.g., a Prusa i3 MK3 printer. This 3D printer has a 250 mm × 210 mm print bed size and a 210 mm build height. Therefore, the 150 mm × 18.8 mm × 450 mm prismatic wing prototype was fragmented into three 150 mm × 18.8 mm × 150 mm and six 75 mm × 18.8 mm × 150 mm parts. These fragmentations introduce additional boundaries, forming the adhesive contact surfaces. The literature [39,40] proved the efficiency of this connection technology. However, the adhesive boundaries alter the total weight of the structure because of the formation of additional contact surfaces and the extra weight of the adhesive. To overcome these unwanted changes, the thickness of the outer shell was reduced to 0.8 mm. At the same time, the 3D honeycomb infill with 2% density was used to ensure the flexural rigidity of the printed structure. Gribniak et al. [102] demonstrated the sufficiency of such infill density, providing the flexural stiffness of the printed parts. Thus, the designed weight of the fragmented prototype (without adhesive) is 207.5 g for the three-fragment slab and 216 g for the six-fragment specimen, i.e., comparable to the single-part sample (213 g, Figure 5a).

The Prusa i3 MK3 printer employed similar settings to those of the single-part PLA samples to fabricate all fragments: extruder temperature = 210 °C; print bed temperature = 60 °C; printing speed = 60 mm/s; layer height = 0.2 mm; nozzle width = 0.4 mm; infill density = 2%; infill pattern was 3D honeycomb. Table 2 lists the manufacturer’s (ColorFabb, Belfeld, The Netherlands) specified mechanical characteristics of the polymeric materials [60,67]. These documents also suggest the printing settings above. Before printing, the filaments were dried for 6 h at 50 °C utilizing the Zyle ZY100FD (Krinona Ltd., Kaunas, Lithuania) dehydrator. This procedure mitigates potential issues arising from moisture evaporation, such as bubbling, popping, or inconsistent extrusion, which could compromise the printed material’s surface finish and structural integrity [103].

Figure 6 shows the prismatic wing prototype’s prismatic fragments fabricated in the vertical position. Figure 6a demonstrates the internal structure of the parts of the three-fragment specimen sliced with the PrusaSlicer 2.3.3 software and prepared for fabrication. Figure 6b,c show the beginning and end of the printing process of these parts. The printing orientation was the adhesive connection variable of the three-fragment specimens—one series of the printed parts had the printing arrangement parallel to the connection joint and another perpendicular to the adhesive joints. Figure 6d shows the parts of the six-fragment bending samples. Six complects of the fragments (two complects for each printing layout’s orientation) were manufactured, as Figure 6 shows, to estimate the scatter in mechanical performance and geometry properties.

The bonding surfaces were gritted with P60 sandpaper to ensure the adhesion of the printed parts. This test program employs two epoxy adhesives for comparison purposes. Adhesive “1” is the two-component epoxy resin EC 152 with hardener W 152 MR (Elantas, Wesel, Germany), mixed in a 10:3 ratio. Adhesive “2” is the Pattex Repair Epoxy Instant Mix Ultra Quick (Henkel, Düsseldorf, Germany).

Figure 7 shows the selected PLA samples; Table 3 describes the bending specimens’ geometry and mechanical characteristics. The geometry was measured using a digital caliper 0–300 mm with a 0.01 mm resolution (Vogel, Kevelaer, Germany); a steel scale was used to measure dimensions exceeding 300 mm. The height (*h*) of all the printed parts was measured in several places: in all corners, in the middle of 75 mm wide fragments, in thirds of 150 mm long sides, and at nine equal distances over 450 mm long sides. So, Table 3 provides the height values averaged from 24 measurement points for single-part samples (S), from 36 points for three-part specimens (F1 and F2), and from 60 points for six-part elements (F3); the width (*b*) and length (*L*) of the sample were averaged from four linear measurements (with corresponding standard deviations). The number after the symbol “±” in this table determines the standard deviation. The camber (*γ*) of the single-part bending samples was also measured at nine equal distances over 450 mm long sides; Table 3 presents only the maximum camber (*γ*_max_). The electronic scale CAS SW-II with a 0.5 g precision (CAS Co., Yangju-Si, Korea) measured the weight (*w_exp_*) of the bending specimens. Table 3 also shows the maximum load (**P***_max_*), the vertical displacement (*u*_P_), and the tensile stresses in the polymeric material (*σ*_P_); both the later parameters correspond to the maximum load. Under the assumption that the printed material behaves elastically until failure and has identical mechanical properties in tension and compression, the following equation determines this stress:(1)$$\sigma =\frac{P\cdot l\cdot h}{8\cdot I},$$
where *l* is the span (400 mm); *I* is the second moment of inertia of the cross-section. In the case of the rectangular empty cross-section, the inertia moment can be calculated as:(2)$$I=\left[b\cdot h^{3}-b^{\prime }\cdot {\left(h^{\prime }\right)}^{3}\right]/12,$$
where *b*′ and *h*′ are the width and height of the space inside the profile. The internal 2% infill (Figure 6a) was neglected when calculating the stress in the fragmented samples.

In addition, Table 3 specifies the bending samples’ modulus of elasticity (*E*). This stress-strain ratio assessment uses Equation (1) to estimate stresses and calculates the corresponding strains as follows:(3)$$\varepsilon =6\cdot u\cdot h/l^{2},$$
where *u* is the vertical displacement at the midspan, this table includes *E* values averaged over the elastic deformation range.

Figure 8 shows the three-point bending sample prepared for the test, which assumes a 400 mm clear span for all samples. A 75 kN capacity electromechanical machine H75KS (Tinius Olsen, Redhill, UK) conducted the bending tests with a ±0.01% position measurement accuracy under displacement control with the 2 mm/min loading velocity; the 4 mm thickness rubber strips protected the polymeric material from the stress concentrations at the support and loading lines. A 50 kN load cell monitored the reaction with 0.5% precision; two 50 mm linear variable displacement transducers (LVDT) measured the vertical displacement at the mid-span with 0.02% accuracy. A workstation computer collected the load cell and LVDT readings every second through the Almemo 2890-9 device (Ahlborn, Holzkirchen, Germany). Table 3 describes the mechanical parameters and classifies the failure mechanisms observed in the bending tests. Still, these characteristics describe the discussion object of Section 3.

#### 2.3.3. Prototyping the Exact Geometry Wing

This case study example employs the CAD model developed in Section 2.2. The wall thickness was reduced from 1.3 mm (Section 2.3) to 1 mm to accommodate the additional weight of the new adhesive connections and preserve the target weight. The 3D printing uses the settings described in Section 2.3.2 for PLA material. Figure 9 defines the 3D printing pathway. The initial fabrication layout did not form the adhesive surfaces—the intention was to connect parts throughout the edges. However, the fabrication process faced limitations because of the stability of slender parts manufactured in the vertical position (Figure 9a), and the CAD model was improved. Figure 10 shows the CAD models of the selected fragments developed with the PrusaSlicer 2.3.3 software. Figure 11 shows the fabricated wing fragments. Table 4 determines the estimated weight of all the wing fragments based on the slicing software and the measurement results; the assessed cost of the polymeric material [60] is 15.73 €.

## 3. Prototyping Results and Discussion

Following the logical structure described in Section 2.3, this section considers three analysis aspects: the printed prototypes’ overall mechanical performance, fracture mechanisms, and the exact wing geometry fabrication. The section also discusses the peculiarities of fabrication (i.e., settings, manufacturing speed, and tolerances) and formulating insights for further study.

### 3.1. The Mechanical Performance of Printed PLA Elements

The comparative analysis of mechanical properties specified in Table 1, Table 2 and Table 3 forms the discussion object of this section. Remarkably, the maximum stresses in 3D printed PLA prototypes (except for the F1-1 sample, Table 3) exceeded 10 MPa, conforming to the minimal values specified in Table 1 (from references [61,63]). Table 2 demonstrates a more substantial difference, identifying a 70 MPa tensile strength for the PLA material considered in the study. However, this seemingly controversial result aligns with the literature [104,105], where pultruded polymeric tubes and standardized tensile samples demonstrated different mechanical resistances. At the same time, the pultruded profiles from the previous studies [104,105] demonstrated less significant discrepancies in strength assessment regarding the standard tensile coupons than 3D printed polymeric specimens (Section 2.1). However, all these experimental outputs proclaim the inconsistency of the standardized tests for predicting the mechanical properties of structural components. In other words, the tensile strength acceptable for homogeneous materials (such as steel) is inadequate for heterogeneous structures fabricated using 3D printing technologies. In the latter case, the load-bearing capacity describes the actual mechanical performance, representing the verification object, as presented in this study. Moreover, the failure of such complex structures is not under tension strength control, as demonstrated by the present tests and previous investigations [96,102,104,105].

The premature failure of the F1-1 sample reveals the importance of the adhesive connection quality, which agrees with the results of the literature [41,42,43]. Even reaching average mechanical resistance (Table 3), the F1-2 specimen’s failure resulted from debonding the adhesive contact at the joint. It may result from the diverse rigidity (expressed in terms of the modulus *E*, Table 3) of the samples with the perpendicular (F1) and parallel (F2) distribution of the printing layers regarding the adhesion joints. Table 2 also confirms the diversity of the modulus of elasticity inside and outside the building plane (*xy*).

As expected from the literature, e.g., [101], LW-PLA substantially reduced the mechanical performance of the test specimens compared to their PLA counterparts. So, Table 3 demonstrates a threefold reduction in the ultimate stress and modulus of elasticity in the single-part sample (S) group. Regarding the geometry characteristics, all flexural samples in this group had comparable height and width; still, the wall thickness (*t*) was different. This difference results from the LW-PLA expansion under 230 °C, which resulted in a 20.6% reduction in the weight (*w_exp_*) regarding PLA. Figure 12 shows images of the PLA and LW-PLA surfaces after a ×25 magnification, which reveal the air bubbles inside the lightweight polymer. Furthermore, the weight reduction of the LW-PLA is well agreed with the target density assumed in Section 2.3.2. Still, the Creality Slicer 4.8.2 software could not capture these changes and regulate the wall thickness.

Regarding fragmentation efficiency, Table 3 does not identify substantial alterations in the mechanical performance, mainly resulting from a relatively small sample size in this case study. However, the load-displacement diagrams shown in Figure 13 reveal the meaningful tendencies. In particular, the single-fragment PLA elements (S-1 and S-2) possess the maximum load-bearing capacity. The reduced stiffness (inclination of the ascending branch) results from the absence of the internal infill and supports of the empty rectangular shape (Figure 4). The six-fragment specimen F3-2 also reached the maximum load-bearing capacity (Table 3) with higher flexural stiffness than S-1 and S-2 elements (Figure 13).

The increase in the mechanical performance of the F3-2 sample results from the emergence of the additional longitudinal adhesion surface (Figure 7 shows the arrangement of adhesive joints in the F3-1, which was the same in the F3-2 sample). The additional longitudinal stiffening plane in F3 specimens also increased the sample weight by 5% regarding the F1 and F2 sample series (Table 3). Alongside the stiffness increase, this longitudinal adhesion joint (Figure 7) has altered the failure mechanism—a load drop terminated the elastic deformation stage at about 6 mm mid-span displacement for both F3 specimens; the differences in the mechanical resistance of nominally identical specimens (F3-1 and F3-2) revealed the unreliable mechanical performance (scattered ultimate resistance) characteristic of all fragmented specimens.

Remarkably, notwithstanding the apparent mechanical performance differences, all wing prototypes resisted the stresses, exceeding the necessary limit of 0.369 MPa (Section 2.3.1). This result supports the viability of the proposed EPO replacement with a 3D printed polymeric shell, and the observed resistance reserve ensures the object for further geometry optimization, reducing the UAV weight and raising the valid payload efficiency. Even LW-PLA specimens ten times overperformed the theoretical stresses (ref. Table 3); still, the low flexural stiffness (inclination of the load-displacement diagrams in Figure 10) will increase the deformations of wings, which may lose aerodynamic shape, and this deformation determines the object of further research.

### 3.2. Fracture Mechanisms of the Bending Elements

Figure 14 shows the typical results of the bending tests of the single-fragment PLA and LW-PLA specimens. Independently of the polymeric material, all single-part prototypes faced brittle failure (designated as “B” in Table 3) because of a fracture of the tension zone. Figure 14a demonstrates that the crack formed at the bottom surface has inclined independently on the printing pathway, proclaiming sufficient bonding strength of the printing layers. After a particular load, the flexural crack changed direction to horizontal, and failure resulted from the brittle crushing of the compressive zone. This crack transformation explains the declination of the load-displacement diagram from the elastic line and further resistance drop, as Figure 13 shows.

Conversely, LW-PLA samples demonstrated a relatively high non-elastic deformation degree until the sudden crack, which followed the 45° inclined printing pathway (Figure 12b) until the specimen separated into two parts; only continuous filming captured the fractured sample image, shown in Figure 14b, because of the sudden fracture process. The LW-PLA density reduction (Table 3) can explain the increase in non-elastic deformations. The weakened interbonding adhesion (regarding ordinary PLA) makes the failure sudden. However, a sufficient mechanical capacity (Section 3.1) makes its applications in manufacturing UAVs possible if deformation limits are satisfied.

Figure 15, Figure 16 and Figure 17 show the results of fragmented specimens. Figure 15a,b demonstrate distinct failure mechanisms characteristic of the bending elements with the perpendicular and parallel distribution of the printing layers regarding the adhesive connection joints. The failure of both specimens with the perpendicular arrangement of the printing layers (F1) resulted from the disintegration of the adhesive joints between the printed fragments (Figure 15a). Thus, these samples did not reach the non-elastic deformation stage, as Figure 9 shows. The relative increase in stiffness of the fragments with the longitudinal arrangement of the printing layers could cause stress concentrations in the joins.

The ductile crushing of the compression zone caused the samples with the printing pathway parallel to the adhesive joints to fail (Figure 15b). Moreover, this ductile failure mechanism ensured remarkable residual resistance while vertical displacements exceeded 30 mm (Figure 9); both F2 specimens demonstrated similar resistance. Therefore, this failure is favorable from the structural safety point of view [103].

Figure 16 shows that the F3 elements demonstrate specific failure with diverse cracking of each printing surface. Fabrication defects (e.g., Figure 17a) could cause such failure. The multiple deformation peaks in Figure 13 could appear for the same reason. At the same time, the adhesive connection flaws (Figure 17b) did not affect the crack formation at the joints. The flowed-away adhesive formed the glassy surface in Figure 17b. Furthermore, the additional longitudinal adhesive joint in these samples (Figure 7) improved the initial (elastic) stiffness of the F3 elements regarding the three-fragment specimen groups F1 and F2, as Figure 13 shows.

The premature failure of the F1-1 specimen (Table 3) has resulted from insufficient bonding quality and is unrepresentative, considering fracture mechanisms. Still, notwithstanding distinct failure mechanisms (Figure 15a,b and Figure 16), F1-2, F2-2, and F3-1 specimens demonstrate equivalent ultimate resistances (Figure 13), which makes them adequate comparison objects. On the one hand, the equivalent ultimate loads (Table 1) reveal nominally identical resistance of the polymeric shell in compression (Figure 15b), tension (Figure 16a), and adhesive bond (Figure 15a). It might be considered an efficient component design. Still, the 10.7–10.9 MPa stresses reached in these samples (Table 3) almost 30 times exceed the theoretical stress value (0.369 MPa, Section 2.3.1), determining the structural optimization space and revealing the preliminary nature of the presented prototyping examples.

### 3.3. Fabrication Example: The Exact Geometry Wing

Section 2.3.3 explains the case study example. Since the wing projection exceeds 1 m (Figure 11b), the fabrication required fragmentation because of the printing space limitation (Figure 11a). This example does not include mechanical tests. Therefore, unlike the simplified bending samples (Section 2.3.2), this prototyping used the two-component fast-gluing cyanoacrylate adhesive Sega Fix 9500 (Sega, Istanbul, Turkey). Analyzing existing prototypes and their fabrication technologies [106] substantiated this adhesive choice. However, the adhesive’s extremely short hardening time substantially complicated the wing assembly, making tailoring the wing parts’ position impossible to ensure a precise geometry.

Figure 18 shows the final view of the wing prototype after adhesively connecting all manufactured fragments. This prototype preserved the CAD shape (Figure 9b) as expected. Table 4 describes the printing characteristics relevant to this context.

The estimated wing material cost of 15.73 € (Section 2.3.3) seems acceptable for manufacturing UAVs. However, the 3D printing process is time-consuming—the total fabrication duration exceeds 64 h (Table 4) if printing a single fragment each time. Thus, combining the printed fragments (e.g., Figure 11a) defines a promising solution. However, it might cause drastic printing errors, as Figure 19a shows. This example results from insufficient bonding of the printed parts with the printing bed. These parts were designed without a solid pad, which caused the error of the printer functioning autonomously. Therefore, it was decided to form solid pads for the elements fabricated in the “vertical” position that solved the issue (e.g., Figure 11a). Another fabrication drawback results from insufficient printing precision, as Figure 19b,c show. This surface “perforation” characteristic of the fragment fabricated in the “horizontal” position (Figure 9a) is due to the discrete layer thickness and the top surface inclination. Figure 19b,c show the connection joints of the fragments fabricated in the horizontal and vertical positions, making the differences in the surface quality apparent.

The calculated and measured weights in Table 4 agree well (the 0.6% printing error is negligible). However, the CAD model outperforms the 764 g mass estimated in the theoretical analysis (Section 2.3.1). This disagreement results from the oversimplified and rough CAD modeling assumptions. This inadequacy aligns with the slicing software limitations mentioned in Section 3.1 (regarding the switch between PLA and LW-PLA settings); it raises the necessity of customized slicing algorithms for advanced materials and structural layouts, as noted in the references [25,91]. This software problem defines the further optimization object.

### 3.4. Further Research

This proof-of-concept study has reached the target, revealing the possibility of fabricating a UAV with PLA materials available on the market [60,67] and desktop 3D printers. Still, the identified safety margins (exceeding 20 times the theoretical resistance capacity) determine the optimizing possibility of the wings and fuselage geometries. Furthermore, the environmental benefits of using a biodegradable material align with the growing emphasis on sustainable engineering practices. This research could pave the way for more advanced and eco-conscious UAV designs, driving broader adoption of additive manufacturing techniques in aerospace and beyond. By utilizing PLA in UAV construction, new paradigms of lightweight, customizable, and rapid prototyping capabilities can be explored that traditional manufacturing methods struggle to achieve. At the same time, PLA-based 3D printing could significantly reduce fabrication costs, allowing rapid prototyping and innovation.

However, prototyping has revealed technical challenges that need to be addressed. These challenges, such as fabrication precision and quality issues (e.g., Figure 17 and Figure 19), underscore the need for control and solutions in industrial applications. The CAD model slicing software, in particular, requires modifications to ensure precise tailoring of the complex geometries of 3D printed components and avoid erroneous settings.

The 3D printing process is time-consuming, as Table 4 shows. Therefore, it is suitable for prototyping purposes (because of fabrication tailoring flexibility). Still, efficient manufacturing requires considering the following aspects:Geometry optimization, including reduction of printing volume (shell thickness), object fragmentation, and parallel fabrication. These means will reduce the fabrication duration and UAV weight;The mass fabrication of optimized geometry shapes must combine different technologies (e.g., continuous filament reinforcement and injection molding), ensuring fabrication quality and speed;Continuous control is essential to prevent fabrication defects (Figure 17a) and terminate manufacturing errors (Figure 19a). This necessity requires the development of online visual monitoring systems, including artificial intelligence (AI) algorithms, which are becoming increasingly crucial in additive manufacturing.

Fabricating the simplified bending samples (Section 2.3.2) employed the adhesive connection technology previously used to form polymeric inserts for structural profiles [96,102]. This technology was acceptable for simple rectangular structural shapes and significant bonding surface areas. However, the adhesive assembly of the wing prototype (Figure 18) was challenging because the adhesive had an extremely short hardening time. Thus, the adhesive choice defines the object for further study and must satisfy the following conditions:The mechanical resistance of the joints must ensure the target load-bearing capacity;The hardening time must ensure the ability to tailor the component position. Alternatively, a scaffold may speed up the assembly process and compensate for the hardening speed;The adhesive should fill empty spaces and voids but not flow out of the gluing space. These conditions determine the adhesive flowability optimization problem.

In addition, future mechanical tests must account for the stress and elevated temperature localization near the servo-equipment mounting places. For instance, electronic speed controllers (ESC) generate substantial heat flow [107] that may reach melting temperatures of a polymeric material (Table 1) and affect its mechanical performance.

This preliminary proof-of-concept UAV fabrication study investigated technological manufacturing feasibility and the material’s mechanical resistance aspects. Therefore, the durability and reliability aspects of UAV exploitation have remained beyond the scope of this research. For instance, the polymeric UAV exposed to direct sunlight may change its geometric shape [82,83] because of PLA’s relatively low glass transition temperature (Table 2); the negative temperatures raise the PLA brittleness [34]; the polymers absorb environmental moisture [82] and change their mechanical performance with age [103], and so on. Not necessarily all these problems are solvable. Still, the corresponding limitations determine the efficient exploitation conditions of the 3D printed polymeric aerial vehicles, which must be explored in future research.

## 4. Conclusions

Additive manufacturing (3D printing) technologies may revolutionize the development of unmanned aerial vehicles (UAVs). Therefore, this study hypothesized that UAVs could be printed using simple desktop printers and polymeric materials. An extensive analysis of the mechanical properties of polymeric materials reported in the literature determined the necessity of prototyping to characterize the 3D printed part’s resistance adequately. So, the research focused on applying polylactic acid (PLA) in manufacturing the flying wing-type UAV. It developed a fabrication concept to re-design UAVs initially produced from a mixture of expanded polystyrene and polyethylene (EPO). This study also investigated the mechanical performance of lightweight PLA (LW-PLA) for comparison purposes, bending the simplified wing prototypes. In addition, this investigation considered the possibility of fragmentation in the manufactured objects because of the limitations of printing space. The experimental program included ten bending samples fabricated from PLA and LW-PLA and three fragmentations (i.e., single-part, three-part, and six-part samples). The mechanical tests revealed the following essential results:Despite differences in mechanical performance, all bending samples demonstrated remarkable resistance to stress, substantially exceeding the theoretical limit of 0.369 MPa. This result provides reassurance about the reliability of the proposed EPO replacement concept with a 3D printed polymeric shell. The 10.7–10.9 MPa stresses reached in PLA samples, almost 30 times exceeding the theoretical limit, determine the structural optimization space for reducing the UAV weight and raising the valid payload efficiency.The potential of LW-PLA for manufacturing UAVs is a hopeful prospect. Raising the extruder temperature from 210 °C to 230 °C for LW-PLA reduced the sample weight by 20.6%, as expected from the literature review, but approximately four times reduced the estimated stresses and modulus of elasticity of the printed material (regarding the ordinary PLA). Still, estimated stresses have exceeded the theoretical limit eight times, which makes LW-PLA acceptable for manufacturing UAVs if deformation limits are satisfied. However, careful optimization of CAD slicing parameters and extrusion temperature is essential to ensure appropriate layer arrangement (overlapping) and particular material expansion to address the relatively weak interlayer bonding strength characteristic for lightweight materials.The bending samples with the perpendicular fragment printing pathway orientation regarding the bending plane demonstrated favorable ductile failure because of the compressive zone crushing. This failure mechanism ensured remarkable residual resistance even for vertical displacements exceeding 30 mm (i.e., 3/40 of the bending span). Thus, this arrangement is recommended for fabricating bending UAV components (i.e., wings).

The case study exemplified fabricating the exact geometry of a fragmented wing prototype, adapting its fabrication to the desktop printer Prusa i3 MK3. The estimated PLA cost of 15.73 € is acceptable for manufacturing UAVs. However, the 3D printing process was time-consuming—the total fabrication duration exceeds 64 h if all the fragments are fabricated separately. Combining the printed fragments may define a promising solution. Unfortunately, it might cause drastic printing errors, as shown in this study. The following means may avoid the printing errors:The slender details, printed in the vertical position (i.e., wing fragments, as recommended in this study), must have a sufficient bonding area with the printing bed. Empty profiles may not ensure reliable contact, so a solid base should be provided. It might be removed after the printing to reduce the structural weight if necessary.Avoid forming surfaces with less than 10% inclination is essential during CAD fragmentation. Due to limited printing resolution, these shapes can lead to unwanted perforations and may require additional surface finishing.The fabrication demonstrated acceptable precision, with a 0.6% error in the weight of the CAD model and fabricated prototype. However, to fully realize the fabrication potential, the existing CAD slicing software needs modification to reduce the weight of aerodynamic profiles.

The fragmented wing prototyping case study also reveals substantial problems that may result from the adhesive choice. In particular, together with the required mechanical resistance, the adhesive’s hardening time must ensure the ability to tailor the component position for the precise shape of the detail; the optimal flowability should ensure the adhesive infills empty spaces and voids but does not flow out of the gluing space.

## Figures and Tables

**Figure 1 polymers-16-02600-f001:**
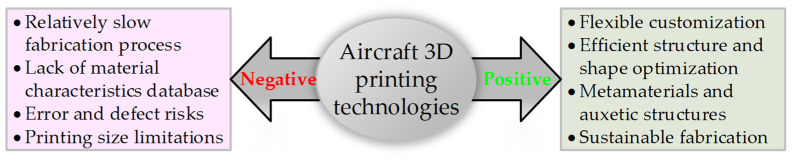
The positive [30,31,32,33,34] and negative [27,30,34,35,36,37] aspects of 3D printing technologies in aircraft engineering.

**Figure 2 polymers-16-02600-f002:**
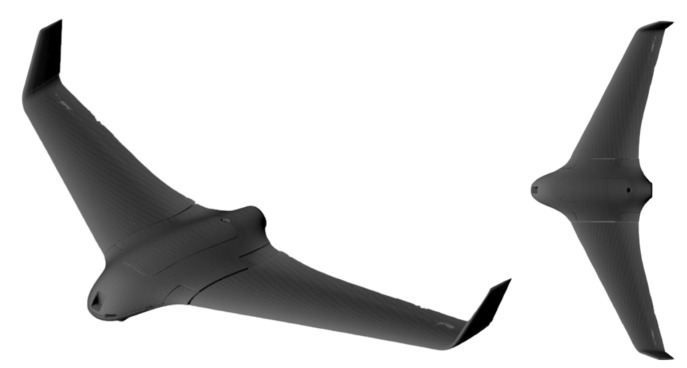
Fabrication object, Skywalker X8: isometric and top views.

**Figure 3 polymers-16-02600-f003:**
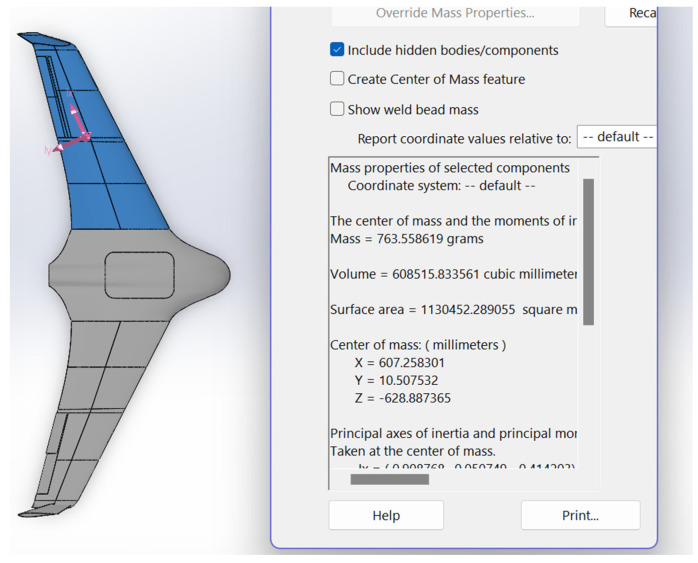
The wing mass assessment iteration result using Solidworks software.

**Figure 4 polymers-16-02600-f004:**
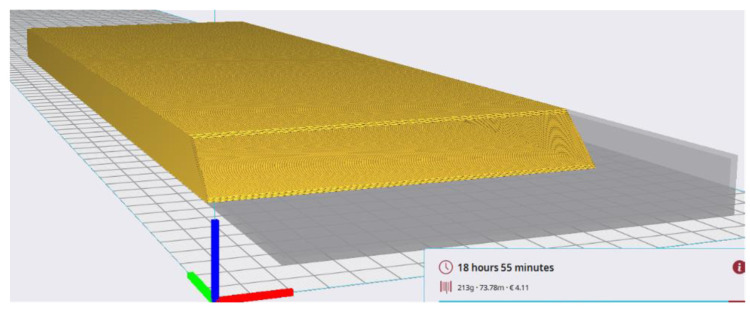
The Creality Slicer software interface of the wing prototype for printing.

**Figure 5 polymers-16-02600-f005:**
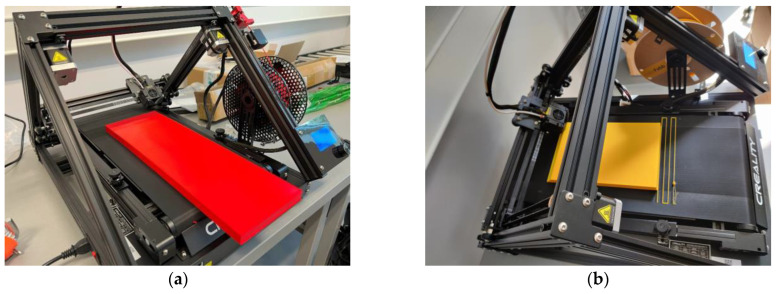
Fabricating the prismatic wing equivalent with the Creality CR-30 3DPrintMill: (**a**) PLA and (**b**) LW-PLA specimens.

**Figure 6 polymers-16-02600-f006:**
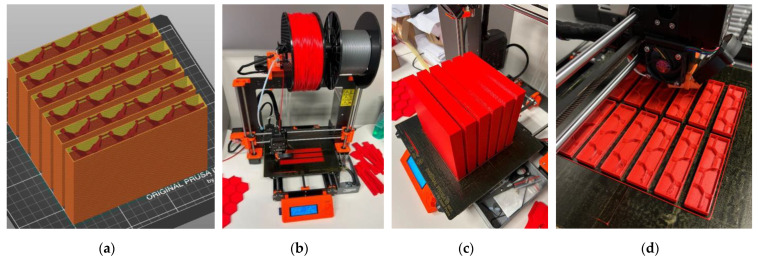
The fragmented wing prototypes: (**a**) CAD model of the 150 mm × 18.8 mm × 450 mm parts sliced with the PrusaSlicer 2.3.3 software; (**b**,**c**) Begin the printing and fabricated 150 mm × 18.8 mm × 450 mm parts; (**d**) Printing 75 mm × 18.8 mm × 150 mm parts.

**Figure 7 polymers-16-02600-f007:**
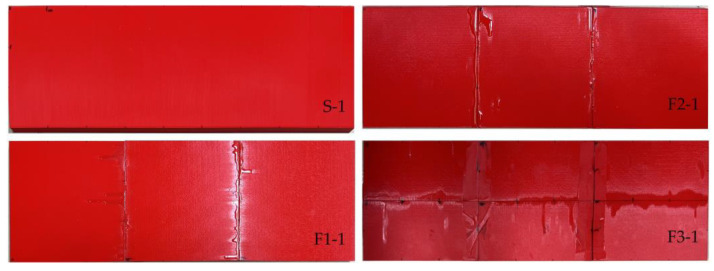
Selected PLA samples printed as a single part (S-1) and fragmented, distributing the printing pathway parallel (F1-1) and perpendicular (F2-1 and F3-1) to the bending plane.

**Figure 8 polymers-16-02600-f008:**
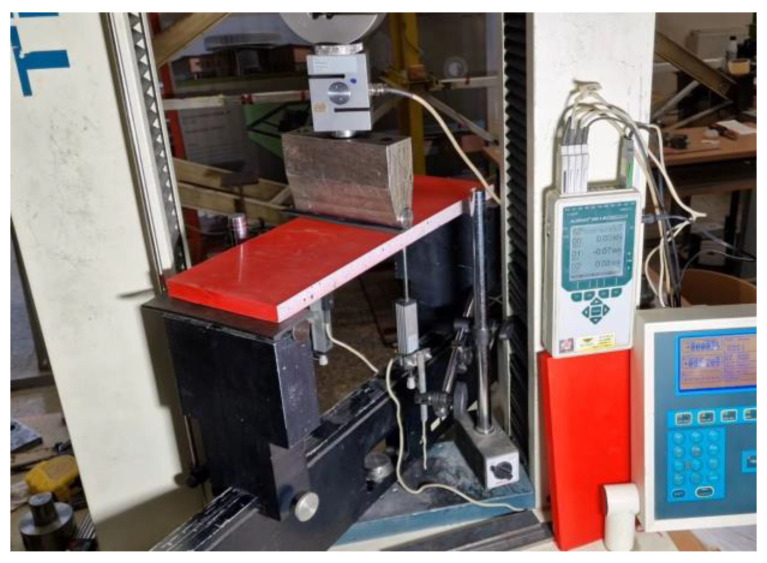
The three-point bending test setup.

**Figure 9 polymers-16-02600-f009:**
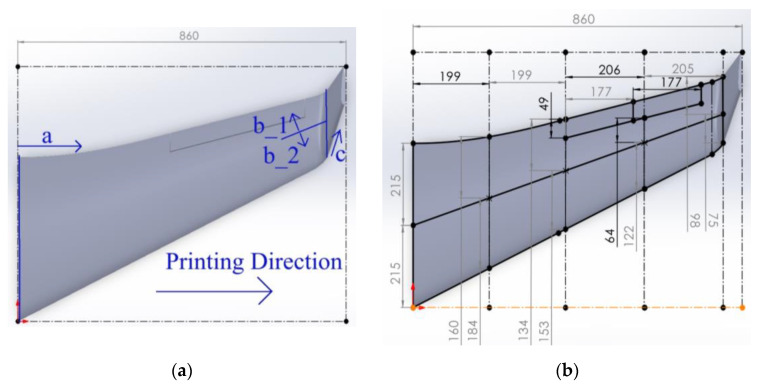
The fabrication concept of the wing: (**a**) Manufacturing pathway; (**b**) Fragmented view.

**Figure 10 polymers-16-02600-f010:**
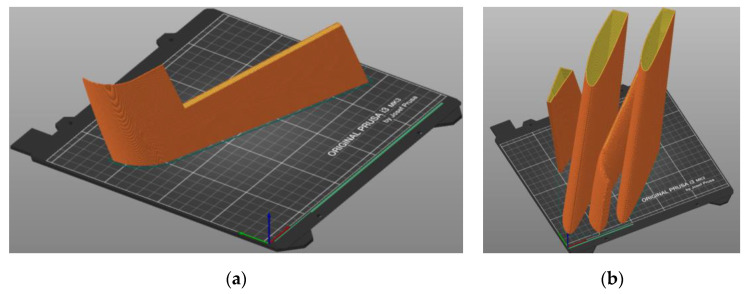
Slicing of the wing: (**a**) A single fragment; (**b**) Several fragments for efficient fabrication.

**Figure 11 polymers-16-02600-f011:**
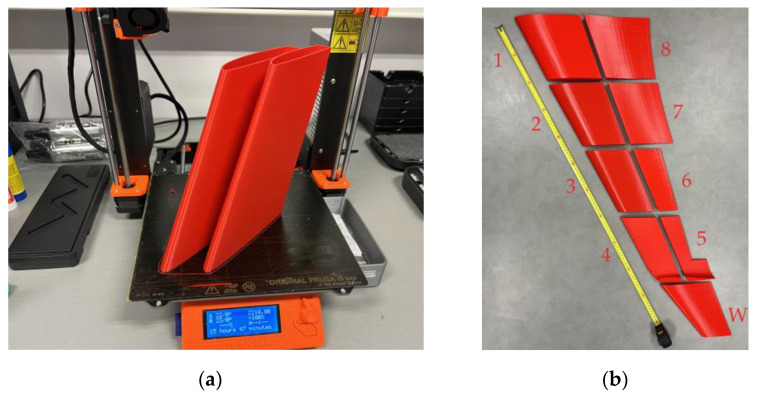
The wing prototyping result: (**a**) Printed fragments; (**b**) All printed fragments.

**Figure 12 polymers-16-02600-f012:**
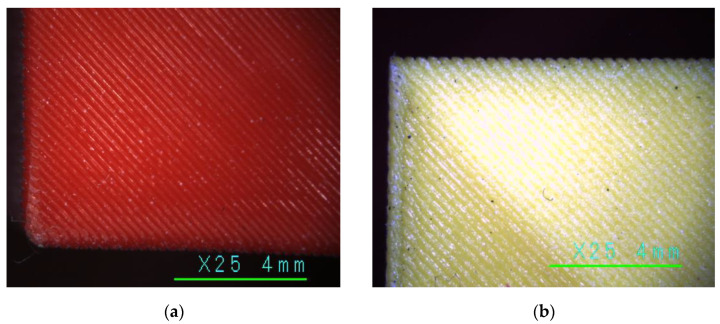
Printed surface (×25 magnification): (**a**) PLA (S-2 sample); (**b**) LW-PLA (S-3LW sample).

**Figure 13 polymers-16-02600-f013:**
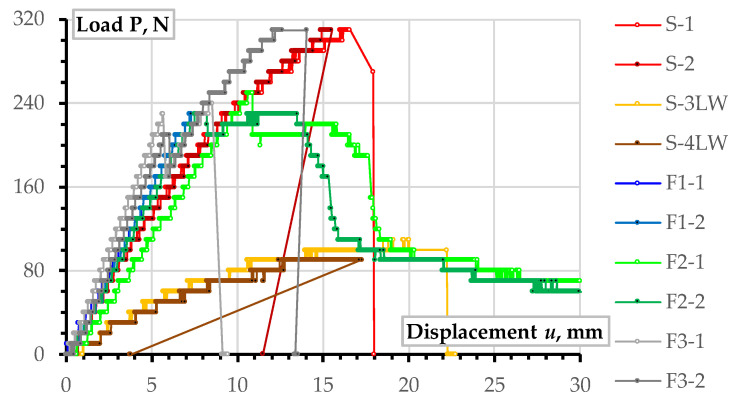
Load-displacement diagrams of all bending specimens.

**Figure 14 polymers-16-02600-f014:**
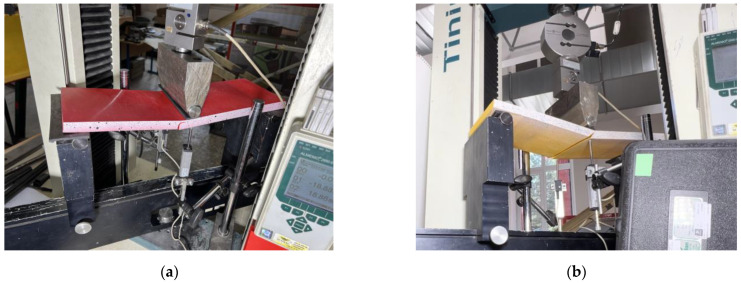
The failure patterns of the single-part specimens: (**a**) PLA (S-1); (**b**) LW-PLA (S-3LW).

**Figure 15 polymers-16-02600-f015:**
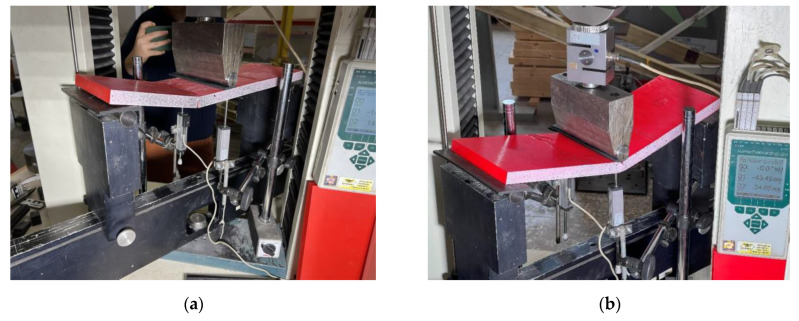
The failure patterns of the three-part bending specimens with different arrangements of the printing pathways regarding the adhesive joints: (**a**) Perpendicular (F1-2); (**b**) Parallel (F2-1).

**Figure 16 polymers-16-02600-f016:**
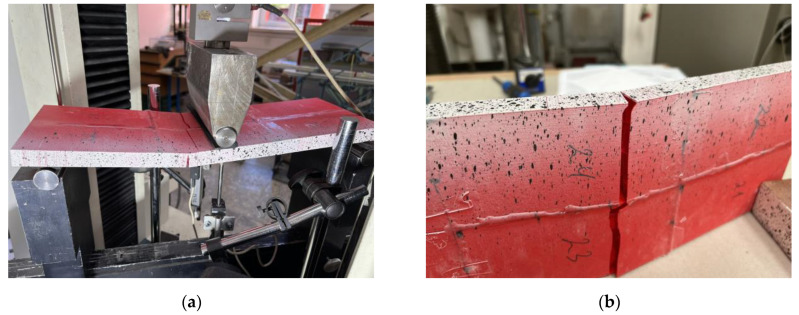
The failure patterns of the six-part specimen F3-2: (**a**) General view; (**b**) Zoomed view.

**Figure 17 polymers-16-02600-f017:**
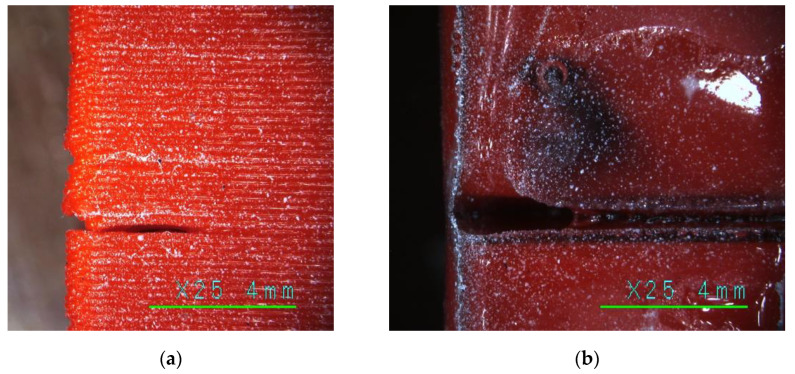
The fabrication defects in the F3-1 specimen (×25 magnification): (**a**) Printing defect; (**b**) Flowing adhesive.

**Figure 18 polymers-16-02600-f018:**
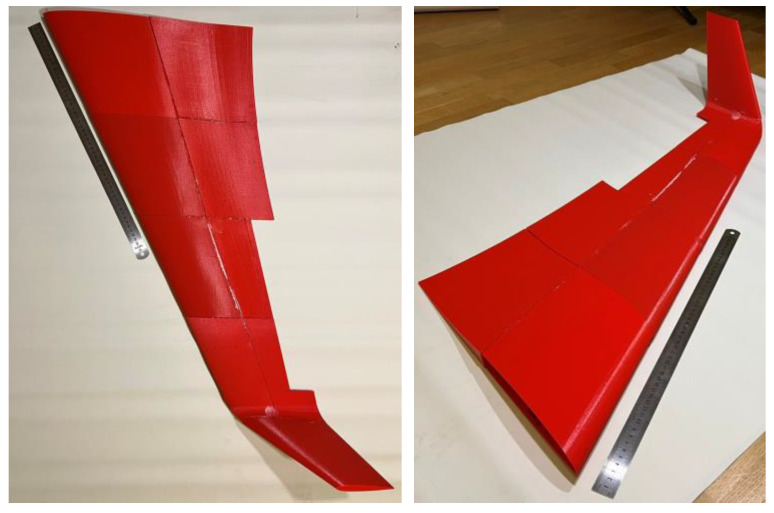
The manufactured wing prototype.

**Figure 19 polymers-16-02600-f019:**
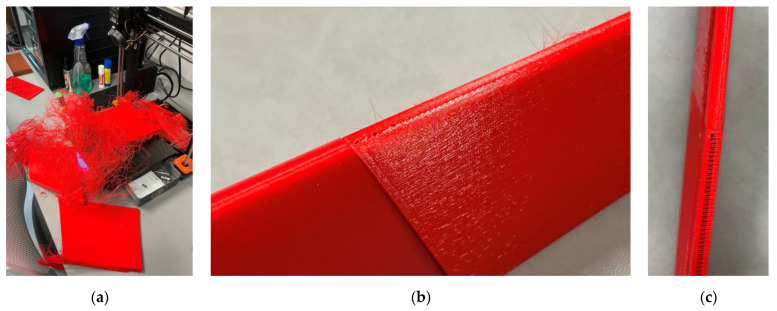
The fabrication gaps: (**a**) Insufficient bonding with the printing bed; (**b**,**c**) The perforation of the front surface of fragment “4” and the empty spaces at the backside of the “5” fragment (refer to Figure 11b for the fragment numbering).

**Table 1 polymers-16-02600-t001:** Characteristics of raw polymeric materials.

Material *	ABS	PLA	LW-PLA	PETG	PEI	PC	PA	ASA
References	[46,47,48,49,50,51,52,53,54,55,56,57,58]	[55,56,57,59,60,61,62,63,64]		[56,65,66,67,68]	[52,53,54,55,69,70,71]	[53,55,72,73,74]	[55,56,57,58,75,76,77,78,79]	[54,55,56,57,80,81,82,83,84]
Advantages	Cheap, lightweight, flexible	Biodegradable, sustainable raw materials		Non-toxic, chemically resistant	Temperature, flammability, and chemically resistant, high strength	Temperature-resistant, high strength	High interlayer adhesion; resistant to abrasion and impact	UV resistant, durable
Weakness	Temperature deformations, toxic gases	Low melting temperature, brittleness	Low melting temperature and modulus of elasticity	High water absorption, hardly printable	Expensive	High printing temperature, deformation instability	High water absorption, deformation instability	Deformation instability, high-precision printing demands
Density [g/cm^3^]	1.04–1.12	1.20–1.26	0.42–1.20 **^◊^**	1.27–1.28	1.17–1.34	1.19–1.20	1.10–1.25	1.05–1.07
Tensile strength [MPa]	32–52	31(11) ^‡^–71	10–43	47–50	54–104	55(19) ^‡^–62	55–63	42–50
Elasticity modulus [GPa]	1.8–2.0	2.3–4.5	0.86–3.4	1.5–1.9	2.1–3.1	2.1–2.4	2.0–3.0	1.6–2.1
Glass transition temp. [°C]	100–112	50–70	55–60	70–75	186–220	145–147	62–79	101–116
Melting temperature [°C]	125–150	150–160	150–160	–	–	228–280	189–194	130–141
Shrinkage ^‡^ [%]	0.5–11	0.3–8.1	−5.8 ^△^–7.8	0.3–12	20	0.8–1.0	0.5–8.1	0.5–0.8
Toxic gas emission	Yes	No	No	No	No	Yes	Yes	Yes
Price [€/kg]	25–33	17–60	35–43	28–30	202–273	43–85	66–71	32–35

* ABS = acrylonitrile butadiene styrene; PLA = polylactic acid; LW-PLA = lightweight PLA; PETG = polyethylene terephthalate glycol; PEI = polyetherimide; PC = polycarbonate; PA = polyamide; ASA = acrylonitrile styrene acrylate. **^◊^** It depends on temperature and foaming factors. **^‡^** It depends on the printing parameters. ^△^ The negative shrinkage corresponds to an expansion deformation.

**Table 2 polymers-16-02600-t002:** Mechanical performance and physical properties of polymeric materials [60,67].

Material	*E* [GPa]		*f_t_* [MPa]		*ε_u_* [%]		*E_f_* [GPa]		*f_r_* [MPa]		*T_g_* [°C]	*T_m_* [°C]
	xy	z	xy	z	xy	z	xy	z	xy	z		
PLA	3.30	3.35	70	71	3.5	3.5	2.40	2.20	97	85	55–60	150–160
LW-PLA	3.35 (0.86) **^◊^**		43 (10) **^◊^**		8.1 (12.8) **^◊^**		–		–		55–60	150–160

**^◊^** The characteristics correspond to the fabrication temperatures at 200 °C (250 °C). Note: *E* = modulus of elasticity; *f_t_* = tensile strength; *ε_u_* = ultimate strain; *E_f_* = flexural modulus of elasticity; *f_r_* = flexural strength; *T_g_* = glass transition temperature; *T_m_* = melting temperature.

**Table 3 polymers-16-02600-t003:** The geometry and mechanical resistance characteristics of the bending specimens.

Sample *	Adhesive	*h* [mm]	*b* [mm]	*L* [mm]	*t* ^‡^ [mm]	*γ_max_* [mm]	*w_exp_* [g]	P*_max_* [N]	*u*_P_ [mm]	*σ*_P_ [MPa]	*E* [GPa]	Failure ^◊^
S-1	–	17.85 ± 0.29	149.96 ± 0.07	447.8 ± 0.5	1.32 ± 0.28	3.99	216.0	310	16.0	11.0	1.27	B
S-2	–	17.86 ± 0.33	149.95 ± 0.19	448.3 ± 0.5	1.36 ± 0.37	2.81	216.0	310	15.0	11.1	1.32	B
S-3LW	–	17.58 ± 0.30	149.97 ± 0.03	447.6 ± 0.5	1.73 ± 0.73	2.15	171.5	110	18.5	3.54	0.356	B
S-4LW	–	17.59 ± 0.27	150.01 ± 0.05	447.8 ± 0.5	1.72 ± 0.70	1.67	171.5	90	12.2	2.92	0.358	B
F1-1	“1”	18.63 ± 0.09	149.54 ± 0.08	449.3 ± 0.3	–	–	206.5	140	4.33	6.65	2.23	J
F1-2	“2”	18.64 ± 0.09	149.34 ± 0.23	449.2 ± 0.3	–	–	210.0	230	7.22	10.7	1.96	J
F2-1	“1”	18.61 ± 0.09	149.72 ± 0.06	448.6 ± 0.4	0.82 ± 0.00	–	208.5	250	10.7	11.6	1.37	D
F2-2	“2”	18.70 ± 0.11	149.73 ± 0.09	448.1 ± 0.4	0.78 ± 0.02	–	211.5	230	7.43	10.9	1.98	D
F3-1	“1”	18.70 ± 0.05	149.69 ± 0.07	450.1 ± 0.2	0.82 ± 0.01	–	233.0	240	8.78	10.7	2.17	B
F3-2	“1”	18.70 ± 0.05	149.67 ± 0.13	450.0 ± 0.2	0.83 ± 0.01	–	227.0	310	11.6	13.7	1.89	B

* The following notation is used: “S” = a single-part sample; “F1” = a three-part fragmented sample with a printing pathway perpendicular to the adhesive joints; “F2” = a three-part sample with a printing pathway parallel to the adhesive joints; “F3” = a six-part fragmented sample with the printing pathway perpendicular to the bending plane; “LW” = fabricated from LW-PLA; the last number determines the sample order within the group. ^‡^ Measured in the failure plane and averaged on four measurements (at the center of each side); the F1 group samples did not reach the fragment fracture. **^◊^** “B” = brittle failure of the tension zone; “D” = ductile failure of the compressive zone; “J” = brittle failure of the adhesion joint.

**Table 4 polymers-16-02600-t004:** Weights and printing time of the wing fragments.

Fragment *	“1”	“2”	“3”	“4”	“5”	“6”	“7”	“8”	“W”	Total
Calculated [g]	125.31	103.80	86.64	69.90	68.09	55.62	143.99	176.89	112.57	942.81
Measured [g]	126.3	102.3	86.0	70.7	68.7	55.9	146.3	177.8	114.0	948.0
Time [h]	10:14	8:27	7:09	5:45	3:45	4:44	8:16	9:52	6:16	64:28

* The fragments’ notation corresponds to Figure 11b.

## Data Availability

The authors will provide the raw data of this work upon a reasonable request.

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
