# Peer review of "Investigating Additive Manufacturing Possibilities for an Unmanned Aerial Vehicle with Polymeric Materials"

_polymers, 2024, doi:10.3390/polym16182600_

Round 1

Reviewer 1 Report

Comments and Suggestions for Authors

This study explores the feasibility of using a simple desktop 3D printer and polymeric materials, specifically polylactic acid (PLA) and lightweight PLA (LW-PLA), to manufacture flying wing-type UAVs. The research focuses on PLA's advantages in non-toxicity, ease of use, and cost, and examines the impact of material density changes on mechanical performance. The study also addresses the issue of object fragmentation due to printing space limitations. Results show significant performance reserves compared to theoretical values, validating the hypothesis and suggesting directions for further optimization. However, some”

1.      On page eight of the article, please clarify that the choice of a colorful shape for the bending test should be explicitly related to the limitations of the 3D printer and the anticipated test results. Additionally, explain the necessity of transforming the wing into a prismatic shape and how this benefits the testing.

2.      On page eight of the article, in the section describing the manufacturing process, please have the author outline the 3D printing specifications: summarize the settings used for PLA and LW-PLA in the paragraph. Regarding sample dimensions and weight, clearly state the samples' dimensions, wall thickness, and weight, as well as the reasons for selecting these specific parameters.

3.      On page fourteen of the article, please have the author explain the impact of adhesive joints. Specifically, detail why the six-fragment sample (F3-2) has greater flexural stiffness and thus higher load-bearing capacity compared to the single-fragment samples.

4.      On page seventeen of the article, the author discusses the differences in deformation between PLA and LW-PLA. Please have the author explain the possible reasons why LW-PLA exhibits significant non-elastic deformation before sudden failure, and how this non-elastic deformation indicates weaker interlayer bonding in LW-PLA compared to PLA.

Comments on the Quality of English Language

polish

Author Response

1) Comment. On page eight of the article, please clarify that the choice of a colorful shape for the bending test should be explicitly related to the limitations of the 3D printer and the anticipated test results. Additionally, explain the necessity of transforming the wing into a prismatic shape and how this benefits the testing.

Answer. The Authors appreciate this note. This proof-of-concept experimental study replaced the cantilever test with the simplified three-point bending test and simplified specimen geometry for two reasons. First, typical wing mechanical tests employ elaborate testing setups because of the variable wing geometry and uniform load distribution. In addition, a limited restraint stiffness is characteristic of printed polymeric structures, which substantially complicates forming the cantilever support, representing the wing and fuselage joint (Figure 3 of the updated manuscript). Second, the existing printing space limits the sample dimensions, which can be fabricated without fragmentation. Therefore, this study transforms the wing into an equivalent rectangular shape, which forms the prismatic sample for the three-point bending test. This test ensures the precise control of the support and loading conditions and captures the deformation and failure of the polymeric element.

Correction in the manuscript. The updated manuscript introduces the corresponding clarification in Lines 307-317.

2) Comment. On page eight of the article, in the section describing the manufacturing process, please have the author outline the 3D printing specifications and summarize the settings used for PLA and LW-PLA in the paragraph. Regarding sample dimensions and weight, clearly state the samples’ dimensions, wall thickness, and weight and the reasons for selecting these parameters.

Answer. The Authors acknowledge this comment. However, merging PLA and LW-PLA in one paragraph may be misleading. Moreover, this study employs two desktop printers to fabricate the test specimens. Thus, the Authors decided to leave these specifications in separate paragraphs (Lines 322–327 and 330–334 of the updated manuscript). At the same time, they found that the description of the Prusa i3 MK3 printer’s settings was unclear. Therefore, the Authors provided these settings on Lines 350–353. At the same time, the Authors do not accept the suggested introduction of the specimen characteristics, which are presented in Table 3 and discussed in detail with the test results in Section 3.

Correction in the manuscript. The Prusa i3 MK3 printer’s settings were described on Lines 350–353.

3) Comment. On page fourteen of the article, please have the author explain the impact of adhesive joints. Specifically, detail why the six-fragment sample (F3-2) has greater flexural stiffness and thus higher load-bearing capacity than the single-fragment samples.

Answer. The Authors appreciate this comment. The increase in the mechanical performance results from the emergence of the additional longitudinal adhesion plane (Figure 7 of the updated manuscript shows the arrangement of adhesive joints in the F3-1, which was the same in the F3-2 sample). The Authors updated Section 3.1 to clarify this issue.

Correction in the manuscript. This new text was added on Lines 514–523 of the updated manuscript.

4) Comment. On page seventeen of the article, the author discusses the differences in deformation between PLA and LW-PLA. Please have the author explain why LW-PLA exhibits significant non-elastic deformation before sudden failure and how this non-elastic deformation indicates weaker interlayer bonding in LW-PLA than in PLA.

Answer. The Authors agree with this comment. The LW-PLA samples demonstrate distinct behavior because of reduced density (which increases the deformability) and weakened interbonding adhesion (which makes the failure sudden).

Correction in the manuscript. The renewed manuscript includes the corresponding explanation in Lines 547–549.

5) Comment on the quality of the English language. Polish the English language.

Answer. The Authors re-revisioned the manuscript and made corrections to the best of their abilities.

Corrections in the revised version: All changes in the text are highlighted in yellow.

Acknowledgment

The authors sincerely thank the Reviewer for sharing his/her time and knowledge and appreciate comments and suggestions that have noticeably improved the manuscript.

Reviewer 2 Report

Comments and Suggestions for Authors

Please find the comments in the attached file. Thanks!

Author Response

Acknowledgment: The Authors express their sincere gratitude to the Reviewer for sharing his/her time and knowledge. They revised the manuscript, accounting for the reviewer’s constructive comments; the yellow color highlights the modifications in the text.

1) Comment. I would suggest adding more quantitative highlights in the abstract.

Answer. The Authors acknowledge this recommendation. However, they want to highlight this research’s “proof-of-concept” nature. Thus, neither printing structures nor settings were optimized to ensure an efficient reserve (safety) of the printed components. So, the results of Table 3 demonstrate the tenfold to thirtyfold overperformance of the experimental resistance (maximum stresses) regarding the theoretical stresses (0.37 MPa) calculated for the wing and fuselage joint in Section 2.3. The latter value was determined, assuming the load coefficient equals unity. Still, the obtained resistance reserve is substantial. The Authors intended to avoid excessive discussions of this issue (because of the apparent necessity of optimization means). Therefore, they used the neutral wording, “significant reserves,” in the Abstract to describe this result, which reflects the nature of this pilot research.

Regarding printing precision, the analysis demonstrates that the printing error did not exceed 1% in most cases, and this inaccuracy depends on the printing settings. Since these settings were not optimized in this study, the Authors intend not to highlight this aspect in the Abstract.

Correction in the manuscript. The Authors highlighted the “proof-of-concept” nature of the research and added the following statement (to point out the limitations of this study): “Focusing on the mechanical resistance, this study ignored rheology and durability issues, which require additional investigations.”

2) Comment. The authors discussed the mechanical properties (three-point bending flexural behavior). Based on the working conditions of aerial vehicles, I think it is critical to add some analysis on surface roughness, impact resistance, and thermal behaviors of the final part. If none of the other performance properties were investigated, it should be stated in the title or scope of the work.

Answer. The Authors appreciate this comment, which is partially related to the abovementioned issue. The eminent Reviewer is undoubtedly correct about the importance of these parameters and effects. Moreover, the polymeric UAV exposed to direct sunlight may change its geometric shape because of PLA’s relatively low glass transition temperature (Table 2); the negative temperatures raise the PLA brittleness; the polymers absorb environmental moisture and change their mechanical performance with age, and so on. Not necessarily all these problems are solvable. Still, the corresponding limitations determine the efficient exploitation conditions of the 3D-printed polymeric aerial vehicles, which must be explored in future research.

This preliminary proof-of-concept UAV fabrication study investigated technological manufacturing feasibility and the material’s mechanical resistance. The mechanical tests proved the proposed fabrication concept to re-design UAVs initially produced from a mixture of expanded polystyrene and polyethylene (EPO). However, the durability and reliability aspects of UAV exploitation have remained beyond the scope of this research. Still, the authors agree with the reviewer’s opinion about this study’s limitations and try to reflect those in the updated manuscript adequately.

Correction in the manuscript. In addition to the modification of the Abstract (see the reply to Comment 1 above), the following improvements were made in the manuscript:

  • The last paragraph of Section 1 was rewritten to clarify the innovative contribution of this research to engineering practice.
  • The last paragraph of Section 3.4 was rewritten to clarify the study’s limitations and formulate further research direction.

3) Comment. The authors printed the PLA parts at 70°C bed temperature. Such temperature is above the glass transition temperature. Did the author consider and analyze the shrinkage and dimensional stability of the part as compared to the design?

Answer. The Authors agree with the importance of the suggested analysis due to the printing errors (e.g., Figure 19a of the updated manuscript). Our research team plans to simulate these printing errors numerically and relate this printing flaw with temperature deformation (shrinkage). However, this analysis defines the object for future study.

On the other hand, the mentioned 70 °C bed temperature was used for LW-PLA to compensate for the printing temperature increase to 230 °C. For comparison, the ordinary PLA was printed using a bed and extruder temperatures of 60 °C and 210 °C. The manufacturers’ documents [60,67] suggested these settings, and this study followed these recommendations. Furthermore, the recommendations by de Freitas & Pegado [101] determine the LW-PLA extrusion temperature to ensure a 20% density reduction. The Authors updated Section 2.3.2 to clarify the choice of the printing parameters.

Correction in the manuscript. Lines 350–355 (of the updated manuscript) clarify the printing parameters’ choice.

4) Comment. Did the authors consider reporting stress vs. strain in Figure 12 instead of Load vs. displacement? The load and displacement are dependent on the dimension of the tested part and the distance between the supporting pillars.

Answer. Equations 1 and 3 assume the elastic behavior of material and identical deformation properties in tension and compression. This approximation is acceptable for homogeneous materials (e.g., steel) but is too rough for 3D-printed polymeric elements. Therefore, these equations are only used to approximate the ultimate stresses σP and modulus of elasticity E in Table 3. Figure 13 (former Figure 12) shows the results of the bending elements of the same geometry suitable for the comparative analysis, including the non-linear deformation range that is inappropriate for the elastic stress analysis.

5) Comment. As shown in Figure 16 a, 3D-printed parts commonly have defects. Did the authors include any replicates in the fabrication and testing? If so, average and standard deviation values should be reported.

Answer. Figure 17a (former Figure 16a) shows that the accidental printing error appeared only in six-part fragmented specimens (F3 specimen series). It might result from reducing the pad area regarding the constant height (compare Figures 6c and 6d of the updated manuscript). Still, this explanation remains a speculation without corresponding deformation analysis. The Authors will consider answering this question in the upcoming research.

Table 3 delivers the standard deviations of the geometric shapes. Other deviations were not estimated statistically in this study.

6) Comment. Black dots can be observed in Figure 15. Could the authors clarify the resource of those black dots?

Answer. The Authors are sorry for this imperfect image resulting from the contrast pattern painting used for the digital image correlation (DIC) analysis. Figure 16a (former Figure 15a) shows this pattern on the front surface of the bending sample exposed to the DIC system. However, the present study did not include these analysis results; therefore, they were not mentioned in the manuscript. The Authors prefer not to discuss the DIC results in this manuscript since they do not extend the present information meaningfully. They also do not have an alternative image to replace the criticized one. So, the authors hope that the above explanation will satisfy the Reviewer and that additional corrections are unneeded.

7) Comment. In Figure 16 b, the authors indicated the flowing adhesives. Did the authors indicate that as the darker regions or a different surface finish compared to Figure 16a? It would be good to have a set of figures showing a closer view of the samples after fabrication (similar to Figure 6, but with a closer view).

Answer. The Authors are sorry for not collecting the requested image set during the tests. The flowed-away adhesive formed the glassy surface in Figure 17b. The darker regions appear because of the marker drawings during the geometry measurement stage. The adhesive dissolved these marks, and the Authors do not have a better-quality image for replacement.

Correction in the manuscript. The comment “The flowed-away adhesive formed the glassy surface in Figure 17b” was added on Line 569 to clarify the differences in the surface treatment.

Reviewer 3 Report

Comments and Suggestions for Authors

Comments on the Quality of English Language

No serious English issue was found. However, overall, the language should be checked to make the flow smoother. 

Author Response

Note: The Authors revised the manuscript, accounting for the reviewer’s constructive comments; the yellow color highlights the modifications in the text.

1) Comment. The authors demonstrated an innovative approach to UAV manufacturing by leveraging 3D printing technologies, which is a promising direction for the future of aerospace engineering. The Introduction provided many details; however, on page 3, lines 101 to 137 should be summarized as a paragraph. Using bullet points is good for slides and reports but not good and professional for a research paper.

Answer. The Authors appreciate the favorable evaluation of the suggested concept and understand the criticism related to the representation of the literature review results. However, the offered modification will distort a systematic view of the benefits and drawbacks of 3D printing technologies collected from several literature sources. Therefore, the authors eliminated the criticized bulleted statement but supplied the text with a new Pros and Cons schematic (Figure 1), which ensures a systematic view of the process complexity.

Correction in the manuscript:

  • The criticized text on Page 3 was rewritten.
  • A new Pros and Cons schematic (Figure 1) has been introduced to ensure a systematic view of the benefits and drawbacks of 3D printing technologies, revealing their complexity.

2) Comment. I understand that the authors wanted to focus on using polymeric materials. However, other materials, such as composites, foam materials, metal alloys, and thermoplastics, could also produce UAVs. Can the authors discuss the advantages and disadvantages of using other materials?

Answer. The Authors appreciate this note. However, this discussion will further extend the lengthy literature review and obscure this work’s essential contribution to engineering practice because of many possible alternatives. At the same time, the Authors understood and accepted the criticism related to this study’s insufficient motivation.

Remarkably, this manuscript investigates the possibility of manufacturing UAVs using a simple desktop 3D printer and polymeric materials and explores the possible stages of computer-aided design (CAD) production of polymeric UAVs. A UAV prototype made from a blend of expanded polystyrene and polyethylene (EPO) is used to determine the aerial vehicle’s surface shape, dimensions, and weight to demonstrate the feasibility of the proposed design concept. This fabrication technology might optimize internal UAV space and structure (Figure 1) and ensure flexible customizing of the CAD model and polymeric UAV. This manufacturing technology also reduces the staffing demands since one technician can operate several printers, and UAV fragmentation ensures the replacement of damaged components and makes the aerial vehicle’s maintenance efficient.

Focusing on the above target, this study carefully selects the printing material and investigates the mechanical resistance of the 3D printed parts, including those fabricated via the fragmentation technique. The focus is on the printing parameters and performance, such as fabrication speed and product tolerances. The mechanical (three-point bending) tests investigate the effects of the printing technology, temperature, speed, layer orientation, and thickness on the load-bearing capacity and stiffness of the printed parts and verify their ability to reach the theoretical component resistance. The corresponding clarification was added in Section 1. The Authors hope this modification clarifies the issue and substantiates the choice of fabrication methodology.

Correction in the manuscript. The last paragraph of Section 1 (Lines 173-189 if the modified manuscript) was rewritten to clarify the choice of fabrication methodology.

3) Comment. Although the authors addressed the issue of printing space limitations through fragmentation, the practical challenges associated with adhesive bonding and structural integrity are not fully resolved. How can the adhesive bonging issue be improved, especially when applying this to large-scale production?

Answer. The Authors appreciate this comment. They applied the adhesive connection technology previously used to form polymeric inserts for structural profiles [96,103]. This technology was acceptable for simple rectangular structural shapes and significant bonding surface areas. However, the present research revealed the substantial complexity of the adhesive bonding procedure, which defines the object for further study. In particular, the choice of the adhesive must ensure the following essential conditions:

  • The mechanical resistance of the joints must ensure the target load-bearing capacity.
  • The hardening time must ensure the ability to tailor the component position. Alternatively, a scaffold may speed up the assembly process and compensate for the hardening speed.
  • The adhesive should have optimal flowability to avoid empty spaces and voids on the one hand and ensure the adhesive does not flow out the gluing space on the other hand.

Correction in the manuscript:

  • Renewed Section 3.3 introduces the details of the adhesive connection procedure of the fragmented wing prototype (Lines 592–598 of the updated manuscript).
  • Updated Section 3.4 formulates the above conditions for the adhesive.
  • Section 4 also includes a new conclusion on the adhesion issues.

4) Comment. Did the authors try to adjust the 3D printing parameters or optimize the 3D printing process?

Answer: The Authors did not optimize the fabrication settings, applying the printing parameters from the manufacturer recommendations [60,67] and previous studies [101–103].

Correction in the manuscript. Section 2.3.2 was modified to clarify this issue.

5-6) Comments. Can the authors provide more characterization information about the PLA and LW-PLA? The glass transition and melting temperatures were there, but how about TGA and rheological properties? The rheology of materials is highly related to the processing.

How about the mechanical properties of printed UAVs at low temperatures? For example, how would the mechanical properties change after exposure to around 4-7 °C?

Answer: The Authors took the liberty of merging these two comments because they are similar in the study’s context. The eminent Reviewer is undoubtedly correct about the importance of these parameters and effects. Moreover, the polymeric UAV exposed to direct sunlight may change its geometric shape because of PLA’s relatively low glass transition temperature (Table 2); the negative temperatures raise the PLA brittleness; the polymers absorb environmental moisture and change their mechanical performance with age, and so on. Not necessarily all these problems are solvable. Still, the corresponding limitations determine the efficient exploitation conditions of the 3D-printed polymeric aerial vehicles, which must be explored in future research.

This preliminary proof-of-concept UAV fabrication study investigated technological manufacturing feasibility and the material’s mechanical resistance. The mechanical tests proved the proposed fabrication concept to re-design UAVs initially produced from a mixture of expanded polystyrene and polyethylene (EPO). However, the durability and reliability aspects of UAV exploitation have remained beyond the scope of this research. Still, the Authors agree with the Reviewer’s opinion about the rheology problems. Therefore, they extended Section 3.4 by formulating the challenging aspects mentioned above for further research.

Correction in the manuscript: The last paragraph of Section 3.4 was rewritten to clarify the raised problems.

7) Comment on the quality of the English language. No serious English issue was found. However, the language should be checked overall to make the flow smoother.

Answer: The Authors wish to express their sincere gratitude to the Reviewer for sharing his/her time and knowledge; comments and suggestions, which noticeably improved the manuscript, are genuinely appreciated. They revised the manuscript again and made corrections to the best of their abilities.

Corrections in the revised version: All changes in the text are highlighted in yellow.

Round 2

Reviewer 1 Report

Comments and Suggestions for Authors

The revised manuscript is suited for publication.

Reviewer 2 Report

Comments and Suggestions for Authors

The reviewer appreciate the authors efforts and edits to address the comment. The quality of the manuscript is improved and proper for publication.